# When two are better than one: Modeling the mechanisms of antibody mixtures

**Tal Einav**[1], **Jesse D. Bloom**[1,2]*

**1** Basic Sciences Division and Computational Biology Program, Fred Hutchinson Cancer Research Center, Seattle, Washington, United States of America, **2** Howard Hughes Medical Institute, Seattle, Washington, United States of America

* jbloom@fredhutch.org

**Data Availability Statement:** All relevant data are within the manuscript and its Supporting Information files.

**Funding:** This work was supported in part by the Mahan Fellowship (https://www.fredhutch.org/en/

## Abstract

It is difficult to predict how antibodies will behave when mixed together, even after each has been independently characterized. Here, we present a statistical mechanical model for the activity of antibody mixtures that accounts for whether pairs of antibodies bind to distinct or overlapping epitopes. This model requires measuring $n$ individual antibodies and their $\frac{n(n-1)}{2}$ pairwise interactions to predict the $2^n$ potential combinations. We apply this model to epidermal growth factor receptor (EGFR) antibodies and find that the activity of antibody mixtures can be predicted without positing synergy at the molecular level. In addition, we demonstrate how the model can be used in reverse, where straightforward experiments measuring the activity of antibody mixtures can be used to infer the molecular interactions between antibodies. Lastly, we generalize this model to analyze engineered multidomain antibodies, where components of different antibodies are tethered together to form novel amalgams, and characterize how well it predicts recently designed influenza antibodies.

## Author summary

With the rise of new antibody combinations in therapeutic regimens, it is important to understand how antibodies work together as well as individually. Here, we investigate the specific case of monoclonal antibodies targeting a cancer-causing receptor or the influenza virus and develop a statistical mechanical framework that predicts the effectiveness of a mixture of antibodies. The power of this model lies in its ability to make a large number of predictions based on a limited amount of data. For example, once 10 antibodies have been individually characterized and their epitopes have been mapped, our model can predict how any of the $2^{10} = 1024$ combinations will behave. This predictive power can aid therapeutic efforts by assessing which combinations of antibodies will elicit the most effective response.

## Introduction

Antibodies can bind with strong affinity and exquisite specificity to a multitude of antigens. Due to their clinical and commercial success, antibodies are one of the largest and fastest

research/divisions/public-health-sciences-division/research/computational-biology/mahan-fellowship.html) from the Fred Hutchinson Cancer Center (TE) and the NIAID of the NIH (National Institute of Allergy and Infectious Diseases: https://www.niaid.nih.gov/) through R01 AI127893 and R01 AI141707 (JBD). JDB is an investigator of the Howard Hughes Medical Institute (https://www.hhmi.org/). The funders had no role in study design, data collection and analysis, decision to publish, or preparation of the manuscript.

**Competing interests:** The authors have declared that no competing interests exist.

growing classes of therapeutic drugs [1]. While most therapies currently use monoclonal antibodies (mAbs), mounting evidence suggests that mixtures of antibodies can lead to better control through improved breadth, potency, and effector functions [2]. There is ample precedent for the idea that combinations of therapeutics can be extremely powerful—for instance, during the past 50 years the monumental triumphs of combination anti-retroviral therapy and chemotherapy cocktails have provided unprecedented control over HIV and multiple types of cancer [3, 4], and in many cases no single drug has emerged with comparable effects. However, it is difficult to predict how antibody mixtures will behave relative to their constitutive parts. Often, the vast number of potential combinations is prohibitively large to systematically test, since both the composition of the mixture and the relative concentration of each component can influence its efficacy [5].

Here, we develop a statistical mechanical model that bridges the gap between how an antibody operates on its own and how it behaves in concert. Specifically, each antibody is characterized by its binding affinity and potency, while its interaction with other antibodies is described by whether its epitope is distinct from or overlaps with theirs. This information enables us to translate the molecular details of how each antibody acts individually into the macroscopic readout of a system's activity in the presence of an arbitrary mixture.

To test the predictive power of our framework, we apply it to a beautiful recent case study of inhibitory antibodies against the epidermal growth factor receptor (EGFR), where 10 antibodies were individually characterized for their ability to inhibit receptor activity and then all possible 2-Ab and 3-Ab mixtures were similarly tested [6]. We demonstrate that our framework can accurately predict the activity of these mixtures based solely on the behaviors of the ten monoclonal antibody as well as their epitope mappings.

Lastly, we generalize our model to predict the potency of engineered multidomain antibodies from their individual components. Specifically, we consider the recent work by Laursen *et al.* where four single-domain antibodies were assayed for their ability to neutralize a panel of influenza strains, and then the potency of constructs comprising 2-4 of these single-domain antibodies were measured [7]. Our generalized model can once again predict the efficacy of the multidomain constructs based upon their constitutive components, once a single fit parameter is inferred to quantify the effects of the linker joining the single-domain antibodies. This enables us to quantitatively ascertain how tethering antibodies enhances the two key features of potency and breadth that are instrumental for designing novel anti-viral therapeutics.

Notably, while we discuss how synergistic interactions could be introduced to increase the model's accuracy at the cost of additional complexity and fit parameters, the success of our simple models suggest that many antibody mixtures function without synergy, and hence that their effects can be computationally predicted to expedite future experiments.

## Results

### Modeling the mechanisms of action for antibody mixtures

Consider a monoclonal antibody that binds to a receptor and inhibits its activity. Two parameters characterize this inhibition: (1) the dissociation constant $K_D$ quantifies an antibody's binding affinity (with a smaller value indicating tighter binding) and (2) the potency $\alpha$ relates the activity when an antibody is bound to the activity in the absence of antibody. A value of $\alpha = 1$ represents an impotent antibody that does not affect activity while $\alpha = 0$ implies that an antibody fully inhibits activity upon binding. Antibodies with an intermediate value ($0 < \alpha < 1$) will partially inhibit receptor activity upon binding [8], whereas antibodies with potency greater than one ($\alpha > 1$) will increase activity upon binding [5]. As derived in S1 Text Section A.1, for an antibody that binds to a single site on a receptor, the activity at a concentration $c$ of

antibody is given by

$$\text{Fractional Activity} = \frac{1 + \alpha \frac{c}{K_D}}{1 + \frac{c}{K_D}}. \tag{1}$$

To characterize a mixture of two antibodies, we not only need their individual dissociation constants and potencies but also require a model for how these antibodies interact. When two antibodies bind to distinct epitopes, the simplest scenario is that their ability to bind and inhibit activity is independent of the presence of the other antibody, and hence that their combined potency when simultaneously bound equals the product of their individual potencies (Fig 1A) [9, 10]. Alternatively, if the two antibodies compete for the same epitope, they cannot both be simultaneously bound (Fig 1B) [11].

We also define the general case of a synergistic interaction where the binding of the first antibody alters the binding or potency of the second antibody (Fig 1C, purple text). This definition encompasses cases where the second antibody binds more tightly ($K_{D,\text{eff}}^{(2)} < K_D^{(2)}$) or more weakly ($K_{D,\text{eff}}^{(2)} > K_D^{(2)}$) in the presence of the first antibody, as well as when the potency of the second antibody may increase ($\alpha_{2,\text{eff}} > \alpha_2$) or decrease ($\alpha_{2,\text{eff}} < \alpha_2$). This also includes cases where two epitopes slightly overlap and partially inhibit one another's binding, and the competitive binding model can be viewed as the extreme limit $K_{D,\text{eff}}^{(2)} \to \infty$ where one antibody infinitely penalizes the binding of the other.

While the synergistic model in Fig 1C has the merit of being highly general, an important feature of the independent and competitive models (Fig 1A and 1B) is that they predict all antibody combinations with few parameters. In both of these latter models, once the $K_D^{(j)}$ and $\alpha_j$ of 10 antibodies are known (which requires $2 \cdot 10$ experiments) and their epitopes are mapped by pairwise interactions ($\frac{10 \cdot 9}{2}$ additional experiments), the potency of all $2^{10} = 1024$ possible mixtures of these antibodies can be predicted without recourse to fitting. In contrast, because the synergistic model allows arbitrary interactions between each combination of antibodies, the behavior of a mixture exhibiting synergy cannot be predicted without actually making a measurement on that combination to quantify the synergy.

For these reasons, in this work we focus on the two cases of independent or competitive binding and show how we can combine both models to transform our molecular understanding of each monoclonal antibody's action into a prediction of the efficacy of an antibody mixture. Deviations from our predictions provide a rigorous way to measure antibody synergy by computing $\frac{K_{D,\text{eff}}^{(2)}}{K_D^{(2)}}$ and $\frac{\alpha_{2,\text{eff}}}{\alpha_2}$.

To mathematize the independent and competitive binding models, we enumerate the possible binding states and compute their relative Boltzmann weights. The fractional activity of each state equals the product of its relative probability and relative activity divided by the sum of all relative probabilities for normalization (see S1 Text Section A.1). When two antibodies bind independently as in Fig 1A, this factors into the form

$$\text{Fractional Activity}_{(\text{distinct epitopes})} = \left( \frac{1 + \alpha_1 \frac{c_1}{K_D^{(1)}}}{1 + \frac{c_1}{K_D^{(1)}}} \right) \left( \frac{1 + \alpha_2 \frac{c_2}{K_D^{(2)}}}{1 + \frac{c_2}{K_D^{(2)}}} \right). \tag{2}$$

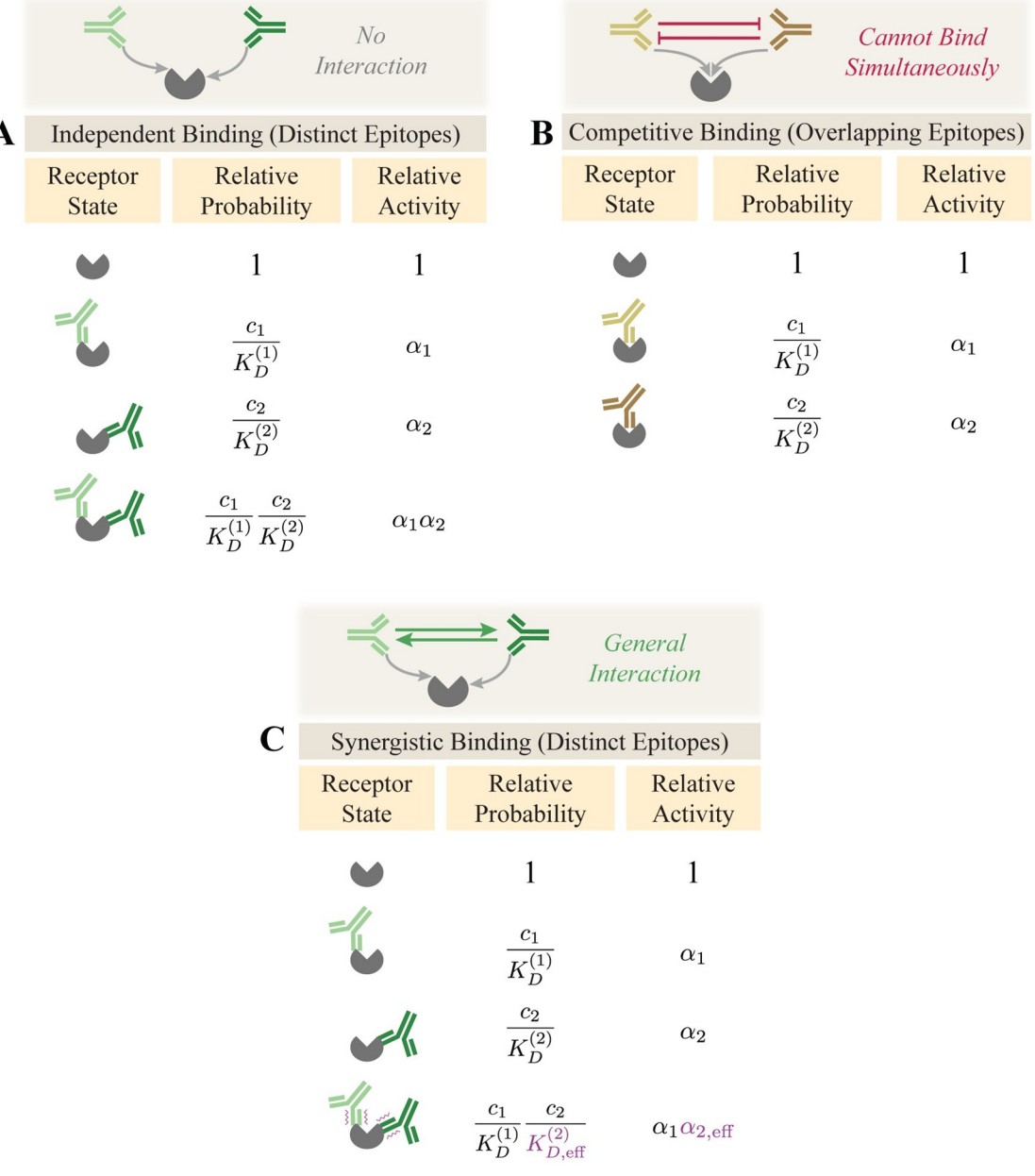

**Fig 1. Binding modes for a 2-Ab mixture.** Two antibodies with concentrations $c_1$ and $c_2$ can bind (A) independently to different epitopes or (B) competitively to the same epitope. (C) Antibodies bind synergistically if either the product of binding affinities ($K_D^{(j)}$) or potencies ($\alpha_j$) are altered when both antibodies bind.

If these two antibodies compete for the same epitope as in Fig 1B, the activity becomes

$$\text{Fractional Activity}_{(\text{overlapping epitopes})} = \frac{1 + \alpha_1 \frac{c_1}{K_D^{(1)}} + \alpha_2 \frac{c_2}{K_D^{(2)}}}{1 + \frac{c_1}{K_D^{(1)}} + \frac{c_2}{K_D^{(2)}}}. \tag{3}$$

These equations are readily extended to mixtures with three or more antibodies (see S1 Text Section A.2).

## Antibody mixtures against EGFR are well characterized using independent and competitive binding models

To test the predictive power of the independent and competitive binding models, we applied them to published experiments on the epidermal growth factor receptor (EGFR) where ten monoclonal antibodies were individually characterized and then the activity of all 165 possible 2-Ab and 3-Ab mixtures was measured [6]. We first use each monoclonal antibody's response to infer its dissociation constant $K_D$ and potency $\alpha$. We then utilize surface plasmon resonance (SPR) measurements to determine which pairs of antibodies bind independently and which compete for the same epitope. These data enable us to use the above framework and predict EGFR activity in the presence of any mixture.

EGFR is a transmembrane protein that activates in the presence of epidermal growth factors. Upon ligand binding, the receptor's intracellular tyrosine kinase domain autophosphorylates which leads to downstream signaling cascades central to cell migration and proliferation. Overexpression of EGFR has been linked to a number of cancers, and decreasing EGFR activity in such tumors by sterically occluding ligand binding has reduced the rate of cancer proliferation [6].

Koefoed *et al.* investigated how a panel of ten monoclonal antibodies inhibit EGFR activity in the human cell line A431NS [6]. They then measured how 1:1 mixtures of two antibodies or 1:1:1 mixtures of three antibodies affect EGFR activity. All measurement were carried out at a total concentration of $2\,\frac{\mu g}{mL}$, implying that each antibody was half as dilute in the 2-Ab mixtures and one-third as dilute in the 3-Ab mixtures relative to the monoclonal antibody measurement.

The 45 possible 2-Ab mixtures (35 binding to distinct epitopes; 10 binding to overlapping epitopes) and the 120 possible 3-Ab mixtures (50 binding to distinct epitopes; 70 binding to overlapping epitopes) were assayed for their ability to inhibit EGFR activity. Fig 2A shows the experimental measurements for mixtures of two antibodies, with the monoclonal antibody measurements shown on the diagonal, the measured activity of 2-Ab mixtures shown on the bottom-left, and the predicted activity on the top-right. The labels on the diagonal entries denote each antibody's binding epitopes inferred through SPR [6], so that antibodies binding to overlapping epitopes are predicted using Eq (3) (pairs within the dashed gray boxes) while mixtures binding to distinct epitopes use Eq (2).

For example, antibodies #1 and #4 bind to distinct epitopes (III/C and III/B, respectively). Hence, the predicted activity of their mixture (0.50) very nearly equals the product of their individual activity ($0.65 \times 0.75 = 0.49$), with the slight deviation arising because each antibody concentration was halved in the mixture ($c_1 = c_2 = 1\,\frac{\mu g}{mL}$ for the 2-Ab mixture characterized by Eq (2), whereas the individual mAbs were measured at $c = 2\,\frac{\mu g}{mL}$ using Eq (1)). This predicted activity roughly approximates the measured value 0.43 of the mixture.

On the other hand, antibodies #1 and #2 bind to the same epitope (III/C), and hence their predicted combined activity (0.67) lies between their individual activities (0.65 and 0.69) since both antibodies compete for the same site. The measured activity of the mixture (0.65) closely matches the prediction of the overlapping epitope model, but is very different than the prediction of 0.45 made by the distinct-binding model.

Fig 2B shows the measured EGFR activity in the presence of all 2-Ab and 3-Ab mixtures is highly correlated with the predicted activity ($R^2 = 0.90$) Notably, the predictions are made solely from the monoclonal antibody data and epitope measurements, and do not involve any fitting of the 2-Ab or 3-Ab measurements. The strong correlation between the predicted and measured activities suggests that EGFR antibody mixtures can be characterized with minimal synergistic effects in either their binding or effector functions. If we did not have the epitope

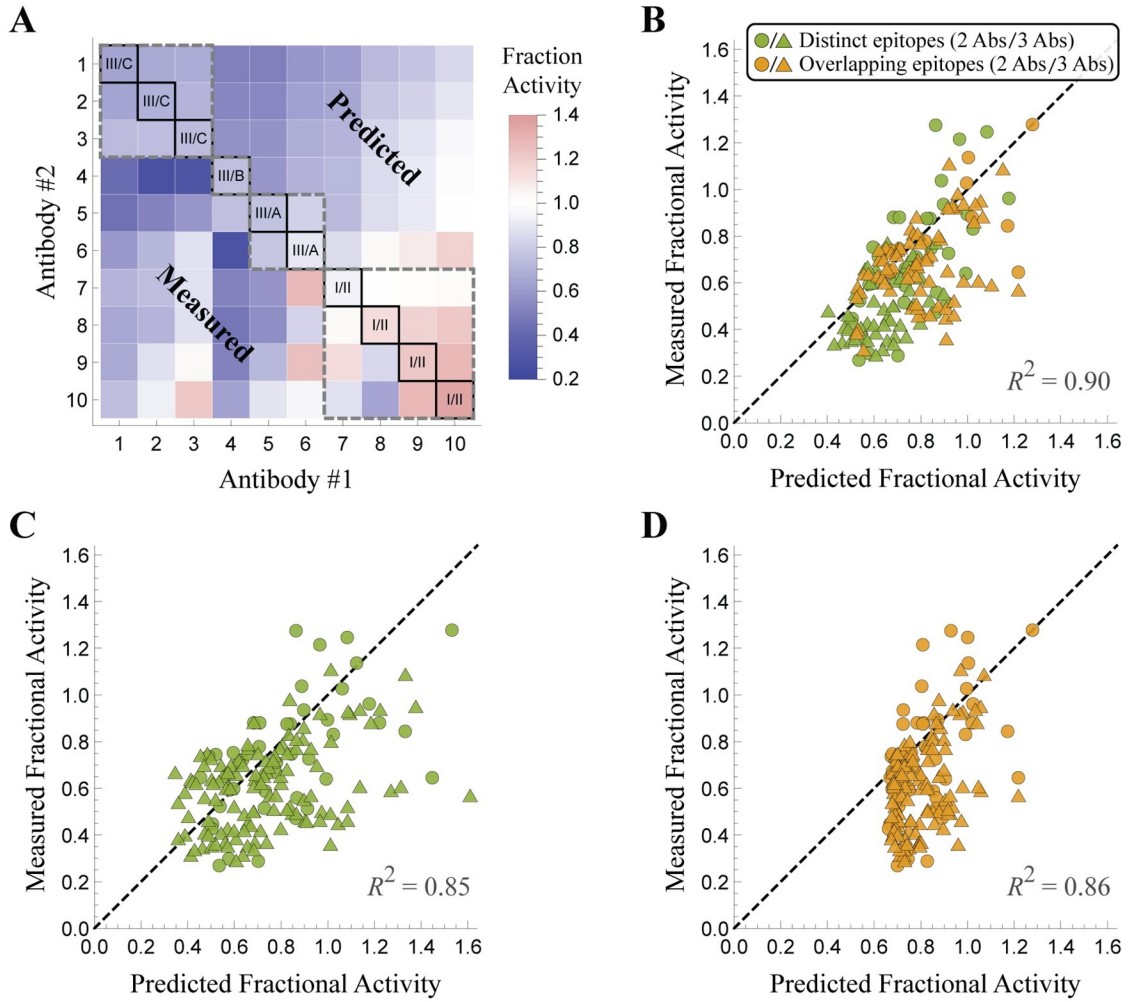

**Fig 2. Predicting how antibody mixtures affect the epidermal growth factor receptor (EGFR).** (A) The fractional activity of EGFR in the presence of monoclonal antibodies (diagonal) together with the measured (bottom-left) and predicted (top-right) activity of all 2-Ab combinations. The dashed gray boxes enclose antibody pairs that compete for the same epitope while all other pairs bind independently. (B) The predicted versus measured fractional activity for all 2-Ab (circles) and 3-Ab mixtures (triangles) using the same epitope mapping as in Panel A inferred by SPR. Without the epitope map, the activity of the mixtures could alternately be predicted by assuming that all antibodies either (C) bind independently or (D) compete for the same epitope; in either case, the resulting predictions fall further from the diagonal line, indicating poorer predictive power. Data was digitized from Table.1 and Fig S1 of Ref. [6].

mapping through SPR and assumed that all antibodies bound to distinct epitopes (Fig 2C, $R^2$ = 0.85) or competed for the same epitope (Fig 2D, $R^2$ = 0.86), the resulting predictions are slightly more scattered from the diagonal, demonstrating that properly acknowledging which pairs of antibodies vie for the same epitope boosts the predictive power of the model.

That said, the predictions incorporating the SPR mapping display a consistent bias towards having a slightly lower measured than predicted activity (Fig E in S1 Text), suggesting that several pairs of antibodies enhance one another's binding affinity or potency. To quantify this effect, we recharacterized the activity from the 2-Ab mixtures using a synergistic model where each $\alpha_{2,\text{eff}}$ is fit to exactly match the data. We find an average value of $\frac{\alpha_{2,\text{eff}}}{\alpha_2} = 0.9$, showing that when pairs of antibodies are simultaneously bound they typically boost their collective inhibitory activity by $\sim 10\%$. This increase in the potency of antibody mixtures could arise from

allosteric interactions where the binding of one antibody stabilizes a binding-favorable confor-
mation for another antibody [12–14].

## Differentiating distinct versus overlapping epitopes using antibody mixtures

In the previous section, we used SPR measurements to quantify which antibodies compete for
overlapping epitopes, thereby permitting us to translate the molecular knowledge of antibody
interactions into a macroscopic quantity of interest, namely, the activity of EGFR. In this sec-
tion, we do the reverse and utilize activity measurements to categorize which subsets of anti-
bodies bind to overlapping epitopes. This method can be applied to model antibody mixtures
in other biological systems where SPR measurements are not readily available.

For the remainder of this section, we ignore the known epitope mappings discerned by
Koefoed *et al.* and consider what mapping best characterizes the data. For example, given the
individual activities of antibody #1 (0.65) and #2 (0.69), the predicted activity of their combi-
nation (at the concentration of $1 \frac{\mu g}{mL}$ for each antibody dictated by the experiments) would be
0.45 if they bind to distinct epitopes and 0.67 if they bind to overlapping epitopes. Since the
measured activity of this mixture was 0.65, it suggests the latter option. We note that such anal-
ysis will work best for potent antibodies (whose individual activity is far from 1), since only in
this regime will the predictions of the distinct versus overlapping models be significantly dif-
ferent. Therefore, the activity measurements of each individual antibody would optimally be
carried out at saturating concentrations (where Eq (1) is as far from 1 as possible).

Proceeding to the other antibodies, we characterize each pair according to whichever
model prediction lies closer to the experimental measurement. To account for experimental
error, we left an antibody pair uncategorized if the two model predictions were too close to
one another (within $4\sigma = 0.16$ where $\sigma$ is the SEM of the measurements) or if the experimental
measurement was close (within $1\sigma$) to the average of the two model predictions (see S1 Text
Section B).

Fig 3A shows how this analysis compares to the experimental measurement inferred by
SPR. While the model predictions are much sparser (with the majority of antibody pairs unca-
tegorized because the two model predictions were too close to one another), the classifications
are nevertheless sufficient to group these antibodies by their epitopes. Antibodies #1-3 all over-
lap with one another (and do not explicitly overlap with any other antibodies) and hence are
assumed to bind one epitope. Antibodies #4 and #5 overlap with each other and form a second
epitope group. Antibody #6 did not explicitly overlap with any other antibody and forms a
third epitope group. Lastly, Antibodies #7-9 all overlapped with #10, and hence these four anti-
bodies bind to a fourth epitope. These four groups are shown by the dashed squares in Fig 3A,
which only disagrees with the epitopes inferred by SPR (shown by the labels on the diagonal
and determined using reference antibodies with known specificities) by claiming that Anti-
bodies #4-5 (rather than #5-6) bind to an overlapping epitope.

These four epitope groups enable us to predict the activity of the 2-Ab and 3-Ab mixtures.
Note that it is not the pairwise classification between two antibodies that determines whether
we apply the distinct or competitive models, but rather the four groupings of antibody epi-
topes. For example, although antibodies #7 and #8 are uncategorized through their 2-Ab mix-
ture, they fall within a single epitope group and hence are considered to bind competitively.
Similarly, antibody #1 and #4 are modeled as binding independently because they belong to
two distinct epitope groups.

Surprisingly, the results shown in Fig 3B have a coefficient of determination $R^2 = 0.90$ that
is on par with the results obtained using the SPR measurements (Fig 2B). Since the inferred

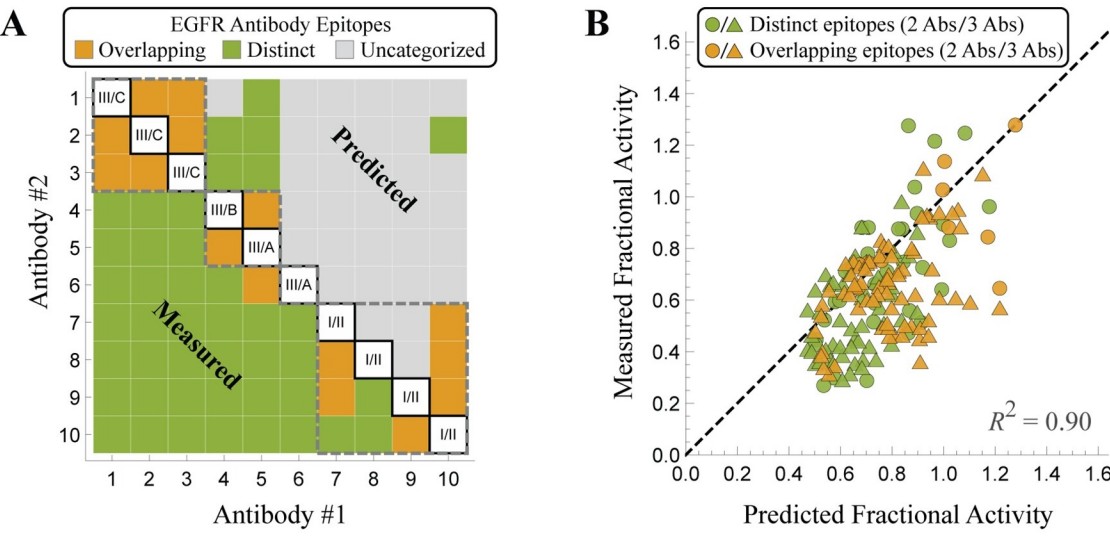

**Fig 3. Classifying antibody epitopes as overlapping or distinct.** (A) Comparing the experimentally measured activity to the overlapping or distinct epitope models enables us to characterize each antibody pair (provided the two models predict sufficiently different activities). The dashed squares represent the minimal epitope groupings inferred from this method. (B) The resulting predictions for the 2-Ab and 3-Ab mixtures have the same predictive power ($R^2 = 0.90$) as a model that relies on epitope groupings given by SPR measurements (Fig 2B).

epitope map relied on the 2-Ab activity data, we compared the predicted activity of the 3-Ab mixtures using the epitopes inferred through SPR with those inferred through the activity data and showed that they are nearly identical ($R^2 = 0.997$, see S1 Text Section A.6). This suggests that there is no loss in the predictive power of the model when an epitope mapping is inferred through activity measurements.

In summary, whether antibodies bind independently or competitively can be determined either: (1) directly through pairwise competition experiments or (2) by analyzing the activity of their 2-Ab mixtures in light of our two models. When this information is combined with the potency and dissociation constant of each antibody, the activity of an arbitrary mixture can be predicted. The Supplementary Information contains programs in both Mathematica and Python that can analyze either form of the pairwise interactions to determine the epitope grouping. If the characteristics of the individual antibodies are also provided, the program can predict the activity of any antibody mixtures at any specified ratio of the constituents.

## Generalizing to models between purely competitive and purely independent binding

Thus far, our model has treated each antibody pair as either binding independently (where the binding of a first antibody has no effect on the second) or binding competitively (where the two antibodies cannot be simultaneously bound). However, SPR experiments measuring the percent decrease of antibody binding in the presence of another blocking antibody can range between or beyond 0% and 100% (Fig 4A). We investigated whether incorporating this more nuanced level of interaction could further refine our characterization of these antibody mixtures.

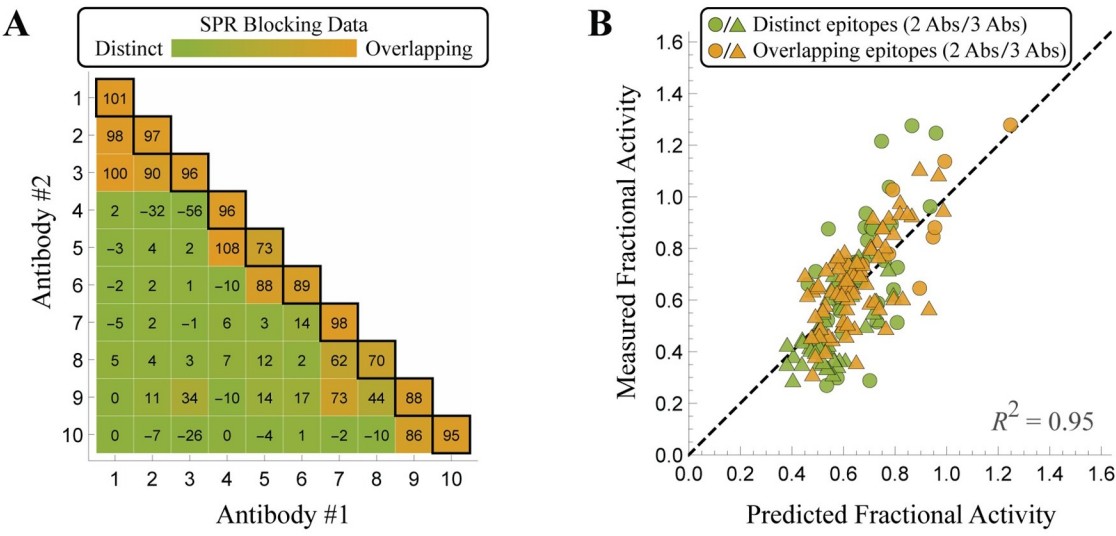

**Fig 4. A continuum model for antibody mixtures.** (A) SPR blocking data from Koefoed *et al.* showing the percentage by which the presence of one antibody inhibits the binding of a second antibody [6]. (B) Characterizing the 2-Ab and 3-Ab mixtures using the continuous binding model Eq (4) that incorporates the continuum of antibody blocking behavior. When antibodies #8-10 are simultaneously bound, their potencies take on the modified values $\alpha_{8,\text{eff}} = 0.60$, $\alpha_{9,\text{eff}} = 1.03$, and $\alpha_{10,\text{eff}} = 0.97$ (see S1 Text Section A.7).

To incorporate partial blocking between antibodies, we generalized the fractional activity of a 2-Ab mixture to

$$\text{Fractional Activity} = \frac{1 + \alpha_1 \frac{c_1}{K_D^{(1)}} + \alpha_2 \frac{c_2}{K_D^{(2)}} + \alpha_1 \alpha_2 f_{12} \frac{c_1}{K_D^{(1)}} \frac{c_2}{K_D^{(2)}}}{1 + \frac{c_1}{K_D^{(1)}} + \frac{c_2}{K_D^{(2)}} + f_{12} \frac{c_1}{K_D^{(1)}} \frac{c_2}{K_D^{(2)}}}, \tag{4}$$

where $f_{12}$ represents the fraction of simultaneous binding for both antibodies (for the $n = 10$ antibodies, these $\frac{n(n-1)}{2}$ parameters are set by $1 - \frac{\% \text{ in Fig 4A}}{100}$ clipped to lie between 0 and 1). In this way, the two antibodies can individually bind to EGFR as dictated by their individual dose-response curves, but the Boltzmann weight of their combined binding is decreased by their partial competition ($K_{D,\text{eff}}^{(2)} = \frac{K_D^{(2)}}{f_{12}}$ in Fig 1C). As expected, Eq (4) reduces to independent binding when two antibodies do not inhibit one another's binding ($f_{12} = 1$) and to competitive binding when one antibody prevents the binding of another ($f_{12} = 0$). For reference, the values on the diagonal represent an antibody competing with itself, and deviations from 100% are likely attributed to noise (e.g. some of the blocking antibody falling off before the test antibody is introduced).

Surprisingly, we found that this continuum model predicted the fractional activity of the 2-Ab and 3-Ab mixtures more poorly ($R^2 = 0.87$; see S1 Text Section A.7) than the original model characterizing every antibody pair as either purely independent or competitive ($R^2 = 0.90$; Fig 2B). More precisely, mixtures containing only Abs #1-7 matched the model predictions far better than mixtures containing Abs #8, #9, or #10 (Fig G in S1 Text). Notably, Abs #8-10 were the only antibodies that individually increased activity ($\alpha > 1$ in the A431NS cell line; Fig B Panel B in S1 Text), yet when mixed with other antibodies they appeared to decrease EGFR activity. For example, while Abs #8 and #10 individually increase activity by 1.14 and 1.35, respectively, their mixture decreases activity to 0.65. This suggested that when Abs #8-10 are simultaneously bound with another antibody, the mechanism of action by which they

increase activity may be disrupted. This idea is corroborated by the observation that antibody mixtures containing Abs #8-10 were systematically higher than the measured activity (Fig G Panel C in S1 Text).

To account for this behavior, we modified the continuum model so that when antibodies #8-10 were simultaneously bound with another antibody, their potency was modified to $\alpha_{j,\text{eff}}$ (in the most general form this adds one parameter for each of the $n = 10$ antibodies; however, we fixed $\alpha_{j,\text{eff}} = \alpha_j$ for antibodies $1 \leq j \leq 7$ whose potency is less than one). These parameter values were inferred from the antibody mixture data to be $\alpha_{8,\text{eff}} = 0.60$, $\alpha_{9,\text{eff}} = 1.03$, and $\alpha_{10,\text{eff}} = 0.97$, and the resulting model predictions are substantially improved ($R^2 = 0.95$; Fig 4B). Hence, although these antibodies increase activity when individually bound to EGFR, when another antibody is simultaneously bound to the receptor they either decrease activity (Ab #8) or keep it essentially constant (Abs #9-10). If Abs #1-7 with individual potency less than 1 are also given $\alpha_{j,\text{eff}}$ values when simultaneously bound with another antibody, only 2/7 acquire an $\alpha_{j,\text{eff}}$ value that substantially differs from their individual potency parameter (see S1 Text Section A.7). This suggests that the mechanism of action for antibodies decreasing EGFR activity is often maintained even when they are simultaneously bound with other antibodies.

In summary, antibodies that individually enhance EGFR activity appear to behave differently (either decreasing activity or leaving it unchanged) when simultaneously bound with another antibody. In contrast, antibodies that individually decrease EGFR activity—likely by blocking ligand binding—will usually act exactly the same when simultaneously bound with another antibody.

## Multidomain antibodies boost breadth and potency via avidity

While the previous sections analyzed combinations of whole, unmodified antibodies, we now extend our framework to connect with the rising tide of engineering efforts that genetically fuse different antibody components to construct multi-domain antibodies [15]. Specifically, we focus our attention on recent work by Laursen *et al.* who isolated single-domain antibodies from llamas immunized with H2 or H7 influenza hemagglutinin (HA) [7]. The four single-domain antibodies isolated in this manner included one antibody that preferentially binds influenza A group 1 strains ($Ab_{A1}$), another that binds influenza A group 2 strains ($Ab_{A2}$), and two antibodies that bind to influenza B strains ($Ab_B^{(1)}$ and $Ab_B^{(2)}$). Fig 5A and 5B shows data from a representative influenza A group 1 strain (blue dot, only bound by the blue $Ab_{A1}$), influenza A group 2 strain (green dot, only bound by the green $Ab_{A2}$), and influenza B strain (gold dot, bound by both of the yellow $Ab_B^{(1)}$ and $Ab_B^{(2)}$ antibodies).

In the contexts of rapidly evolving pathogens such as influenza, two important characteristics of antibodies are their potency and breadth. Potency is measured by the inhibitory concentration $IC_{50}$ at which 50% of a virus is neutralized, where a smaller $IC_{50}$ represents a better antibody. Breadth is a measure of how many strains are susceptible to an antibody.

In an effort to improve the potency and breadth of their antibodies, Laursen *et al.* tethered together different domains using a flexible amino acid linker (right-most columns of Fig 5A and 5B) and tested them against a panel of influenza strains. To make contact with these multi-domain constructs, consider a concentration $c$ of the tethered antibody $Ab_{A1}$–$Ab_{A2}$. As derived in S1 Text Section C.1, the $Ab_{A1}$ or $Ab_{A2}$ portions of the antibody will neutralize the virus with relative probability $\frac{c}{IC_{50,A1}}$ or $\frac{c}{IC_{50,A2}}$, respectively, relative to the unbound HA state. Although neutralization is mediated by antibody binding, the two quantities may or may not be proportional [16–18], and hence the $IC_{50}$s in the denominators need not equal the antibody dissociation constants.

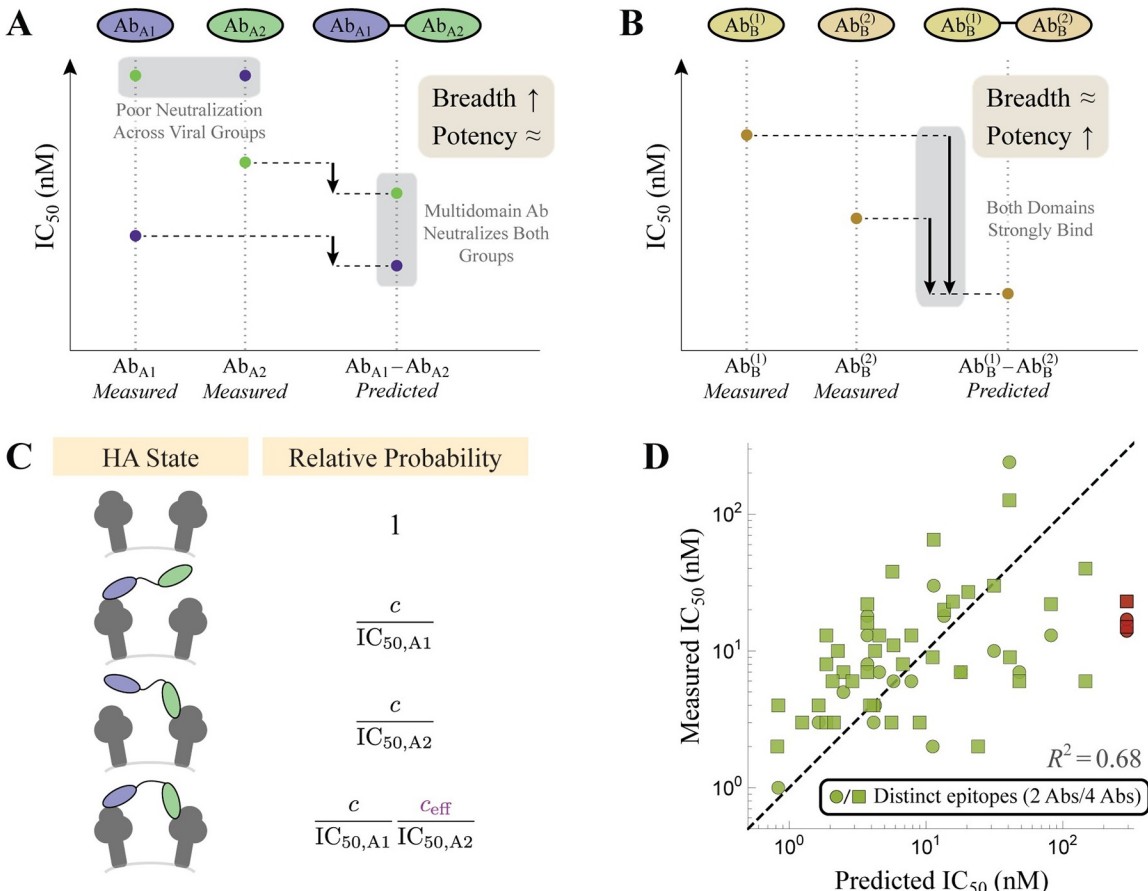

**Fig 5. Tethering influenza antibodies increases breadth and potency.** (A) The influenza A antibodies $Ab_{A1}$ and $Ab_{A2}$ were tethered together to form $Ab_{A1}-Ab_{A2}$ while (B) two influenza B antibodies formed $Ab_B^{(1)}-Ab_B^{(2)}$. Representative data shown for an influenza A group 1 (blue), influenza A group 2 (green), and influenza B (gold) strains. Strong potency is marked by a small $IC_{50}$ while large breadth implies that multiple strains are controlled by an antibody. (C) Representative states of HA and their corresponding Boltzmann weights for multidomain antibodies, where crosslinking between adjacent spikes boosts neutralization via avidity ($c_{eff}$ = 1400 nM in Eq (6)). (D) Theoretical predictions of the potency of all multidomain antibodies versus their measured values. The red points denote two outlier influenza strains discussed in the text that are not neutralized by $Ab_{A1}$ or $Ab_{A2}$ individually but are highly neutralized by their combination.

Laursen *et al.* determined that their tethered constructs cannot intra-spike crosslink two binding sites on a single HA trimer, but they can inter-spike crosslink adjacent HA [7]. The linker connecting the two antibody domains facilitates such crosslinking, since when one domain is bound the other domain is confined to a smaller volume around its potential binding sites. This effect can be quantified by stating that the second domain has an effective concentration $c_{eff}$ (Fig 5C, purple), making the relative probability of the doubly bound state $\frac{c}{IC_{50,A1}}\frac{c_{eff}}{IC_{50,A2}}$ (see S1 Text Section C.1). Therefore, the fraction of virus neutralized by two tethered antibody domains is given by

$$\text{Fraction Neutralized} = \frac{\frac{c}{IC_{50,A1}} + \frac{c}{IC_{50,A2}} + \frac{c}{IC_{50,A1}}\frac{c_{eff}}{IC_{50,A2}}}{1 + \frac{c}{IC_{50,A1}} + \frac{c}{IC_{50,A2}} + \frac{c}{IC_{50,A1}}\frac{c_{eff}}{IC_{50,A2}}}. \tag{5}$$

Note that this equation assumes that influenza virus is fully neutralized at saturating concentrations of antibody ($\alpha = 0$ in Eq (1), with Fraction Neutralized analogous to $1 -$ Fractional Activity).

The $IC_{50}$ of the tethered construct is defined as the concentration $c$ at which half of the virus is neutralized, which can be solved to yield

$$IC_{50,A1-A2} = \frac{IC_{50,A1}\,IC_{50,A2}}{c_{eff} + IC_{50,A1} + IC_{50,A2}}, \tag{6}$$

with an analogous expression holding for the $Ab_B^{(1)}$-$Ab_B^{(2)}$ construct. Using the measured $IC_{50}$s of $Ab_{A1}$-$Ab_{A2}$ and $Ab_B^{(1)}$-$Ab_B^{(2)}$ against the various influenza strains, we can infer the value of the single parameter $c_{eff} = 1400$ nM. This result is both physically meaningful and biologically actionable, as it enables us to predict the $IC_{50}$ of the tethered multidomain antibodies against the entire panel of influenza strains. Fig 6A and 6B compares the resulting predictions to the experimental measurements, where plot markers linked by horizontal line segments indicate a close match between the predicted and measured values.

The two tethered antibodies display unique trends that arise from their compositions. Since the two domains in $Ab_{A1}-Ab_{A2}$ bind nearly complementary strains, the tethered construct will increase breadth (since this multidomain antibodies can now bind to both group 1 and group 2 strains) but will only marginally improve potency. Mathematically, if $Ab_{A1}$ binds tightly to an influenza A group 1 strain while $Ab_{A2}$ binds weakly to this same strain ($IC_{50,A2} \to \infty$), their tethered construct has an $IC_{50,A1-A2} \approx IC_{50,A1}$. Said another way, $Ab_{A1}-Ab_{A2}$ should be approximately as potent as a mixture of the individual untethered antibodies $Ab_{A1}$ and $Ab_{A2}$. Note that since the experiments could not accurately measure weak binding ($>1000$ nM), the predicted $IC_{50}$s for the multidomain antibodies represent a lower bound.

On the other hand, tethering the two influenza B antibodies yields a marked improvement in potency over either individual antibody, since both domains can bind to any influenza B strain and boost neutralization via avidity. The process of engineering a multivalent interaction is reminiscent of engineered bispecific IgG [15], and adding additional domains could yield further enhancement in potency, provided that all domains can simultaneously bind.

While the model is able to characterize the majority of tethered antibodies, it also highlights some of the outliers in the data. For example, the H3N2 strains A/Panama/2007/99 and A/Wisconsin/67/05 were poorly neutralized by either $Ab_{A1}$ or $Ab_{A2}$ ($IC_{50} \geq 1000$ nM), but the tethered construct exhibited an $IC_{50} = 14$ nM and $IC_{50} = 17$ nM, respectively, far more potent than the 300 nM lower limit predicted for both viruses (red circles in Fig 5D and red lines in Fig 6A). Interestingly, Laursen *et al.* found that mixing the individual, untethered antibodies $Ab_{A1}$ and $Ab_{A2}$ also resulted in shockingly poor neutralization ($IC_{50} \geq 1000$ nM), suggesting that the tether is responsible for the increase in potency [7]. From the vantage of our quantitative model, this outlier cries out for further investigation.

To further boost neutralization, Laursen *et al.* created two additional constructs that combined all four antibody domains, the first being the linear chain ($Ab_{A1}$-$Ab_{A2}$-$Ab_B^{(1)}$-$Ab_B^{(2)}$). Since the influenza A antibodies do not bind the influenza B strains (and vise versa), this construct should have the same $IC_{50}$ as $Ab_{A1}-Ab_{A2}$ for the influenza A strains and as $Ab_B^{(1)}$-$Ab_B^{(2)}$ for the influenza B strains, as was found experimentally (compare the *Predicted* columns in Fig 6A–6C). For example, the two H3N2 strains (A/Panama/2007/99 and A/Wisconsin/67/05) were again found to have measured $IC_{50}$s (15 nM and 23 nM) far smaller than their predicted lower bound of 300 nM (red squares in Fig 5D, red lines in Fig 6C).

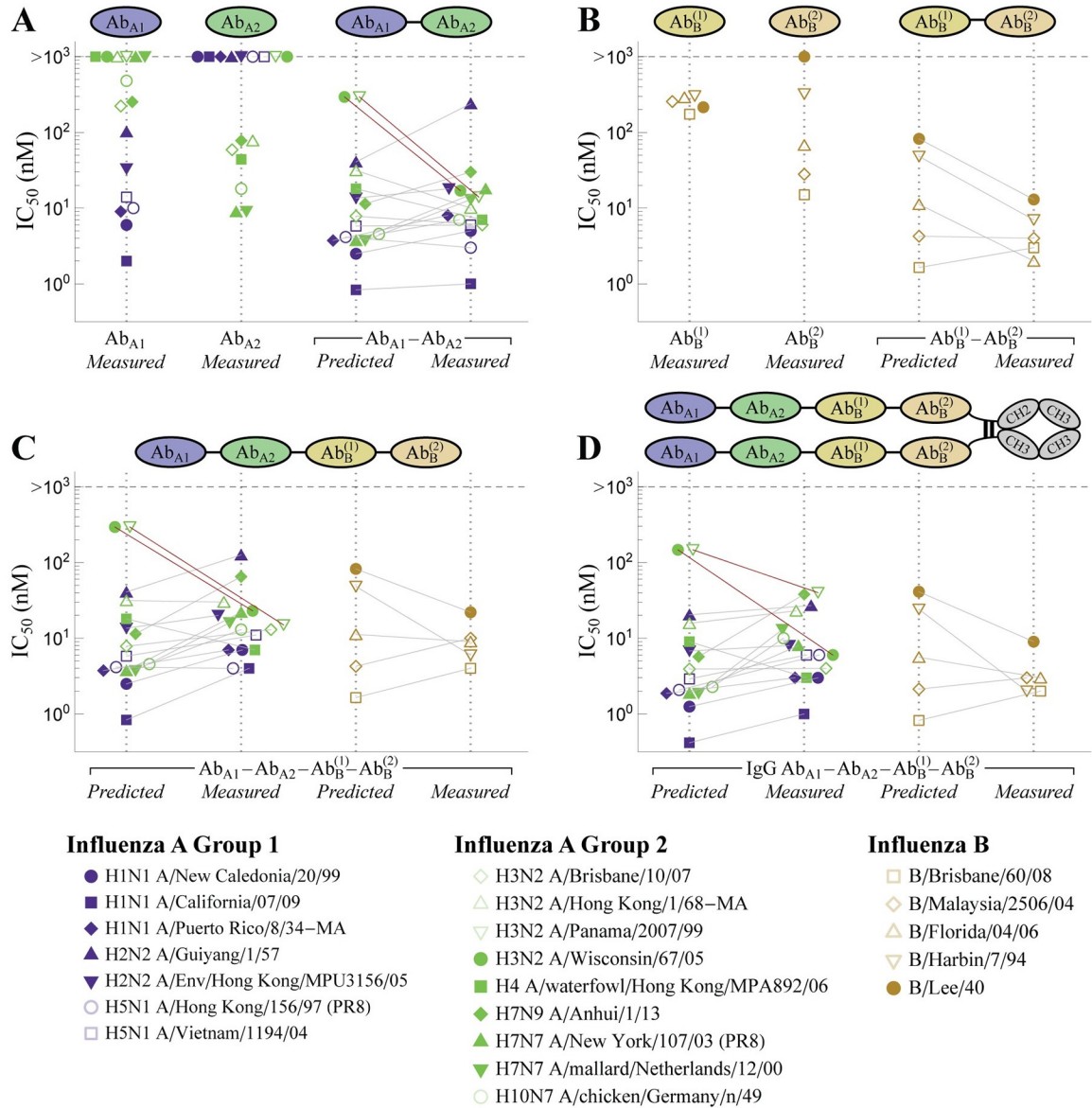

**Fig 6. Neutralization of multidomain antibodies.** (A,B) The potency of the 2-Ab constructs and their constitutive antibodies against a panel of influenza strains. $Ab_{A1}$ primarily binds influenza A group 1 (blue), $Ab_{A2}$ to influenza A group 2 (green), and the two $Ab_B$ antibodies to influenza B strains (gold). (C) All four antibodies were tethered to form the linear chain $Ab_{A1}$-$Ab_{A2}$-$Ab_B^{(1)}$-$Ab_B^{(2)}$ and (D) two copies of this chain were placed on an IgG backbone. The model suggests that the two arms of the IgG are not capable of simultaneously binding a virion. Red lines indicate two outlier influenza strains discussed in the text that are not neutralized by $Ab_{A1}$ or $Ab_{A2}$ individually but are highly neutralized by their combination. Data was digitized from Figs 1 and 3 of Ref [7].

A second construct containing all four antibody domains attached two copies of $Ab_{A1}$-$Ab_{A2}$-$Ab_B^{(1)}$-$Ab_B^{(2)}$ through an IgG backbone (Fig 6D). Since the identical domains in both arms of this construct should be able to simultaneously bind, the new antibody should markedly boost potency through avidity. Surprisingly, the neutralization of this final construct was well characterized as half the $IC_{50}$ of an individual $Ab_{A1}$-$Ab_{A2}$-$Ab_B^{(1)}$-$Ab_B^{(2)}$, suggesting that there was no noticeable avidity and that the increase in neutralization only arose from having twice as many antibody domains. As above, this intriguing result presents an

opportunity to both quantitatively check experimental results and to advocate for future studies. In this particular instance, it suggests that the IgG backbone used was not able to simultaneously bind with both arms. If a different multivalent scaffold (perhaps with greater flexibility or with longer linkers) enabled bivalent binding of both linear antibody chains, it could potentially increase the neutralization of this construct by 100-fold as seen in the influenza B constructs.

## Discussion

In this work, we developed a statistical mechanical model that predicts the collective efficacy of an antibody mixture whose constituents are assumed to bind to a single site on a receptor. Each antibody is first individually characterized by its ability to bind the receptor (through its dissociation constant $K_D$) and inhibit activity (via its potency $\alpha$) as per Eq (1). Importantly, this implies that the activity of each monoclonal antibody must be measured at a minimum of two concentrations in order to infer both parameters, and additional measurements would further refine these parameter values and the corresponding model predictions.

After each antibody is individually characterized, the activity of a combination of antibodies will depend upon whether they bind independently to distinct epitopes or compete for overlapping epitopes. Theoretical models often assume for simplicity that all antibodies bind independently, and in the contexts where this constraint can be experimentally imposed such models can accurately predict the effectiveness of antibody mixtures [9]. Yet when the antibody epitopes are unknown or when a large number of antibodies are combined, it is likely that some subset of antibodies will compete with each other while others will bind independently, which will give rise to a markedly different response. Our model generalized these previous results to account for antibody mixtures where arbitrary subsets can bind independently or competitively (Eqs (2) and (3), S1 Text Section A.2).

We showed that in the context of the EGFR receptor, where every pairwise interaction was measured using surface plasmon resonance, our model is better able to predict the efficacy of all 2-Ab and 3-Ab mixtures than a model that assumes all antibodies bind independently or competitively (Fig 2). This suggest that mixtures of antibodies do not exhibit large synergistic effects. More generally, similar models in the contexts of anti-cancer drug cocktails and anti-HIV antibody mixtures also found that the majority of cases that were described as synergistic could instead be characterized by an independent binding model [9, 10]. This raises the possibility that synergy is more the exception then the norm, and hence that simple models can computationally explore the full design space of antibody combinations.

While it is often straightforward to measure the efficacy of $n$ individual antibodies, it is more challenging to quantify all $\frac{n(n+1)}{2}$ pairwise interactions and determine which antibodies bind independently and which compete for an overlapping epitope. We demonstrated that after each antibody is individually characterized, our model can be applied in reverse by using the activity of 2-Ab mixtures to classify whether antibodies compete or bind independently (Fig 3). Surprisingly, while the resulting categorizations were much sparser than the direct SPR measurements, the classifications produced by this method predicted the efficacy of antibody combinations with an $R^2 = 0.90$, comparable to the predictions made using the complete SPR results (Fig 2B). This suggests that key features of how antibodies interact on a molecular level can be indirectly inferred from simple activity measurements of antibody combinations.

Although these models classified antibody epitopes as either distinct or overlapping, SPR measurements indicate that there is a continuum of possible interactions. Surprisingly, when we generalized our binding model to explore this broader class of behaviors, we found that it resulted in poorer model predictions (S1 Text Section A.7). More specifically, the three

antibodies (#8-10) that individually increased EGFR activity seemed to decrease activity when simultaneously bound with another antibody, representing an important form of synergy that was neglected in the previous simpler models. To account for this behavior, we introduced $n = 10$ effective potency parameters $\alpha_{j,\text{eff}}$ (one per antibody) to quantify the potency of each antibody when simultaneously bound with another antibody. Rather than fitting each of these parameters to the data, we found that fixing $\alpha_{j,\text{eff}} = \alpha_j$ for antibodies satisfying $1 \leq j \leq 7$ (whose individual potency was less than one) and only fitting $\alpha_{j,\text{eff}}$ for the three antibodies that individually increased fractional activity led to a substantial improvement in the model (Fig 4). The effective potency of all three antibodies was reduced by at least 15%, corroborating the notion that when simultaneously bound with another antibody, their effect on EGFR activity may differ from when these antibodies are individually bound (S1 Text Section A.7).

Modern bioengineering has opened up a new avenue of mixing antibodies by genetically fusing different components to construct multi-domain antibodies [15]. Such antibodies can harness multivalent interactions to greatly increase binding avidity by over 100-fold (as seen by the $IC_{50}$s of the A/Wisconsin/67/05 and B/Harbin/7/94 strains in Fig 6). For such constructs, the composition of the linker can heavily influence the ability to multivalently bind and neutralize a virus [18, 19], although Laursen *et al.* surprisingly found little variation when they modified the length of their amino acid linker (see Table S11 in Ref [7]). Another curious feature of their system was that placing their linear 4-domain antibody (Fig 6C) on an IgG backbone (Fig 6D) only resulted in a 2-fold decrease in $IC_{50}$, suggesting that the two "arms" of the IgG could not simultaneously bind. We would expect that a different backbone that allows both arms to simultaneously bind would markedly increase the neutralization potency of this construct. In this way, quantitatively modeling these multidomain antibodies can guide experimental efforts to design more potent constructs.

## Methods

### Models of EGFR antibody binding

Antibody mixtures from Ref. [6] were first characterized using a binding model (Eqs (2) and (3) for 2-Ab mixtures; Eqs (S6)-(S8) for 3-Ab mixtures) where every antibody pair either binds independently or dependently. Model parameters are given in Fig B Panel B of S1 Text.

Antibody epitopes were determined using SPR blocking data (Fig 3A, bottom-left), with two antibodies categorized as overlapping if the average of the two antibody measurements (with preincubation by either antibody) were > 50% and as distinct if the average was < 50% (exact values given in Fig 4A). The reverse process using the antibody mixture data to determine whether antibodies have distinct or overlapping epitopes is described in S1 Text Section B.

A continuum model that incorporates partial competition between each pair of antibodies (Eq (S9) for 2-Ab mixtures; Eq (S10) for 3-Ab mixtures) is described in S1 Text Section A.7. In this model, antibodies are allowed to partially compete for the same epitope (Fig 1C) with the amount of competition dictated by SPR blocking data (Fig 4A).

### Models for influenza multidomain antibodies

Influenza multidomain antibodies from Ref. [7] were characterized using a neutralization model derived in S1 Text Section C.1, Eq (S17). Combining a binding model that accounts for the avidity of the multiple domains together with a sigmoidal relationship between binding and neutralization [17], we derive an expression for the neutralization of these multidomain antibodies. Assuming a Hill coefficient of 1 between binding and neutralization, this model is

identical to the distinct binding model used for EGFR antibodies (1 − Fractional Activity in Eq (2)) with potency $\alpha_j = 0$ for each antibody domain, $K_D \to \text{IC}_{50}$, and $c_2 \to c_{\text{eff}}$.

## Goodness of fit

The coefficient of determination used to quantify how well the theoretical predictions matched the experimental measurements (relative to the dashed line $y = x$ in Figs 2B–2D, 3B, 4B and 5D) was calculated using

$$R^2 = 1 - \frac{\sum_{j=1}^{n} \left( y_{\text{measured}}^{(j)} - y_{\text{predicted}}^{(j)} \right)^2}{\sum_{j=1}^{n} \left( y_{\text{data}}^{(j)} \right)^2} \tag{7}$$

where $y_{\text{measured}}$ and $y_{\text{predicted}}$ represent a vector of the measured and predicted activities for the $n$ mixtures analyzed. In Fig 5D, we computed the $R^2$ of $\log_{10}(\text{activity})$ to prevent the largest activities from dominating the result (since the $\text{IC}_{50}$ values span multiple decades).

## Experimental data

All data are available in the Supporting Information S1 File. Data from the EGFR antibody mixtures was obtained by digitizing Ref [6] Fig S1 using WebPlotDigitizer [20]. Data for the influenza multidomain antibodies was obtained from the authors of Ref [7]. The original nomenclature for the antibodies used in Koefoed *et al.* and Laursen *et al.* are given in Table A in S1 Text.

## Supporting information

**S1 Text. Aforementioned derivations and discussions.**
(PDF)

**S1 File. Contains all of the data analyzed in this work.**
(ZIP)

**S2 File. A supplementary Mathematica notebook contains the data analyzed in this work, recreates all plots shown, and contains a program that takes in the activity of individual antibodies and their pairwise interactions and predicts the activity of all mixtures.**
(ZIP)

**S3 File. A supplementary Python notebook analogous to S2 File.**
(ZIP)

## Acknowledgments

We thank Joost Kolkman and Nick Laursen for useful discussions on their multidomain antibodies and for sharing their data. We thank Mikkel Pedersen and Rob Phillips for their insights on modeling antibody mixtures.

## Author Contributions

**Conceptualization:** Tal Einav, Jesse D. Bloom.

**Investigation:** Tal Einav, Jesse D. Bloom.

**Writing – original draft:** Tal Einav, Jesse D. Bloom.

**Writing – review & editing:** Tal Einav, Jesse D. Bloom.

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
