## [Decision Letter · Decision Letter 0]

21 Dec 2019

Dear Dr Bloom,

Thank you very much for submitting your manuscript 'When two are better than one: Modeling the mechanisms of antibody mixtures' for review by PLOS Computational Biology. Your manuscript has been fully evaluated by the PLOS Computational Biology editorial team and in this case also by independent peer reviewers. The reviewers appreciated the attention to an important problem, but raised some substantial concerns about the manuscript as it currently stands. While your manuscript cannot be accepted in its present form, we are willing to consider a revised version in which the issues raised by the reviewers have been adequately addressed. We cannot, of course, promise publication at that time.

Sincerely,

Rustom Antia

Associate Editor

PLOS Computational Biology

Alice McHardy

Deputy Editor

PLOS Computational Biology

[LINK]

The reviewers, and particular reviewer 3 have raised substantial concerns which I need to be explicitly addressed if this paper is suitable for PLOS computational biology.

Reviewer's Responses to Questions

**Comments to the Authors:**

Reviewer #1: A cocktail of antibodies can be a superior treatment in comparison to single monoclonal antibodies, but understanding the mechanisms behind this observation is still missing. The manuscript by Einav et al. proposes a framework for prediction of the activity of the mixture of antibodies based on measured parameters of individual antibodies (such as affinity, activity and epitope mapping), and, thus, the manuscript addresses an important question. Authors used several experimental data sets to verify their model predictions, and their analysis reports interesting results.

Although the manuscript is well written, and I enjoyed reading it, it has some limitations that need to be addressed before its publication:

1. The model introduced parameter “alpha” (in lines 48-49 and S1 text) ranging 0-1, where alpha=1 corresponds to no effect and alpha=0 implies complete inhibition of receptor activity upon binding by antibody. Underlying model assumes the monotonic relationship between binding of EGFR receptor by growth factors and activation of the cells, where no activation will be seen if all EGFR receptors are blocked by antibodies.

The problem appears when parameter “alpha” is estimated in Figure S2. Parameter “alpha”<1 for all 10 antibodies was derived for HN5 line, but >1 for three antibodies for A431NS. “alpha”>1 corresponds to the case where blocking of all EGFR receptors with antibodies will lead to maximal cell growth. Comparison of “alpha” values for HN5 and A431NS for these three antibodies suggests that the key hidden assumption of the model about monotonic relationship between binding of EGFR receptors by growth factors and activation of the cells is somehow not true (at least for these three antibodies and used concentrations).

One way to deal with it is to analyze only 7 antibodies. Will it lead to better overall predictions in Figure 2C? Currently, model predicts higher fractional activity than measured for many antibody mixtures. Will it still be true if only 7 antibodies and their mixtures will be used in the analysis?

2. What is R2 in Figure 2B-D correspond to? Is it for comparison of the model predictions to the data (shown line y=x) or for the overall best fit of the data (y=ax+b, which is a different case and, in this case, it will be interesting to see parameters a and b)?

3. please include the original experimental data as requested by the journal policy (for example, measured values for fractional activity data for Figure 2B) in the supplemental information

Minor

1. p.4 line 72 “and their epitopes are mapped” add “by pairwise interactions”

2. suggestion for Figure 2B -- to change “2/3Abs” (it reads “two thirds of Abs” which is confusing) to “2Abs/3Abs” and additionally define circles and triangles somewhere in the text

3. few typos: see lines 146, 220 in main text and first sentence on p. S2

Reviewer #2: This paper addresses the timely topic of complex antibody-antigen interactions mediated by two or more antibodies simultaneously present. The authors take a clear biophysical modelling approach to predict the binding activity of antibody mixtures to EGFR and show that the model using epitope mapping data indeed improves upon models assuming only independent or competitive binding. I have the following observations:

1. The basic modelling approach is not new. A model introduced by Wagh et al., PLoS Pathogens, 2016 also considers competitive and independent binding and seems to have large overlap. Can the authors comment on how their model is different?

2. The authors mention possible avenues to extend the basic model for antibody binding. In particular, synergy between antibodies can boost their activity with 10% and note that a more 'nuanced level of interaction' between antibodies could lead to a more precise prediction of dissociation constants. An extension of the binding model is, however, not pursued. In view of comment 1, I feel that an attempt to include 'a continuum of possible interactions' to the binding model could greatly improve the novelty of the modelling approach taken by the authors.

3. Another important point of the manuscript is that multidomain antibodies boost breadth and potency via avidity. However, avidity and boost free energies are not clearly integrated into the basic biophysical model and the corresponding discussion remains somewhat imprecise. I would suggest to integrate the notion of avidity already at the level of equations (2) and (3).

4. The authors motivate their study of antibody mixtures by stating that 'mounting evidence suggests that mixtures of antibodies can behave in fundamentally different ways [2]'. Does this refer to the competitive or independent binding dynamics of mixtures? The clinical trials that are mentioned in this reference only investigate bnAb-combinations with non-overlapping target sites, such that independent binding dynamics apply.

5. From Fig. 5 on the prediction of IC50's of multi-domain antibodies, it is difficult to quantify the prediction of the model. Can the authors quantify how well you can predict the IC50 of the multi domain antibodies, and how much the model improves on an uninformed binding model?

Reviewer #3: Motivated by the increasing popularity of combination antibody therapy, the authors developed an equilibrium binding model to describe the effect of an antibody mixture on the activity of a multi-epitope protein, and applied this model to published data sets on a cancer-related receptor and influenza viral proteins. By assuming either independent or exclusive binding of antibodies, the authors calculated the effect of antibody combinations, using measurements of individual antibodies and pairwise interactions or epitope mapping as inputs. They further considered engineered multi-domain antibodies and compared the predicted and measured potency.

My comments are as follows:

1. The authors claimed that ‘the power of this model lies in its ability to make a large number of predictions based on a limited amount of data. For example, once 10 antibodies have been individually characterized, our model can predict how any of the 2^10=1024 combinations will behave.’ This is not true, given that either pairwise measurements or information of epitope mapping has to be provided. More importantly, the assumption of either independent or exclusive binding seems not properly validated; e.g. as shown in Fig. 2A, prediction and measurement do not show a good match. In addition, the measured activity is systematically lower than the predicted value. Both observations indicate potential importance of synergistic effect which is neglected in the analysis. Synergistic effects (e.g. through allostery) could be even more prominent in compliant protein molecules, which might restrict the generality of this model. This issue should be directly addressed.

2. The authors stated that deviations from the predictions provide a rigorous way to measure antibody synergy, but no analysis is provided to support this statement. It would be more convincing to directly demonstrate this measure for the datasets used. This may offer a metric for the predictive power of this simplified model.

3. ‘This predictive power paves the way to expedite the design of future therapeutics’. This is a superfluous statement. I urge the authors to discuss in concrete terms what general design principles are learnt from this study.

4. In Fig. 3A, it seems that the model fails to predict the epitope of most of the pairs, because two models predict very similar activities. And yet, this lack of epitope information doesn’t impair the correlation between measured and predicted activity (Fig. 3B). Isn’t this showing the model is not well constrained and lacking predictive power?

5. The outliers in Fig. 5 are glaring. Analysis with synergistic effect should be added to show whether remedy can be achieved.

6. As the author pointed out, SPR measurements indicate that there is a continuum of possible interactions, how this aspect shall be accounted for while still maintaining the simplicity and hence scalability of the model should be discussed in depth.

7. The main text is hard to follow without frequently referring to the SI text. The Methods section in the main text should be expanded to include the essential information and formula for each subsection in Results.

**Have all data underlying the figures and results presented in the manuscript been provided?**

Reviewer #1: No: commented on it to the authors

Reviewer #2: Yes

Reviewer #3: Yes

PLOS authors have the option to publish the peer review history of their article (what does this mean?). If published, this will include your full peer review and any attached files.

Reviewer #1: No

Reviewer #2: No

Reviewer #3: No

---

## [Decision Letter · Decision Letter 1]

28 Mar 2020

Very nice paper in a very interesting area .... please take care of the typos indicated.  Best regards, rustom

Dear Dr Bloom,

We are pleased to inform you that your manuscript 'When two are better than one: Modeling the mechanisms of antibody mixtures' has been provisionally accepted for publication in PLOS Computational Biology.

Best regards,

Rustom Antia

Associate Editor

PLOS Computational Biology

Alice McHardy

Deputy Editor

PLOS Computational Biology

Reviewer's Responses to Questions

**Comments to the Authors:**

Reviewer #1: The authors did significant revisions of the manuscript in response to Reviewers comments.

All my concerns have been addressed. It is a well-written, well-organized paper, and I recommend it for publication.

Reviewer #2: The authors have addressed all of my points in a satisfactory manner. I believe that the extensive revision has significantly improved the manuscript and I am particularly impressed by the new section on the continuum model, in which the authors show that antibodies can behave differently when they are simultaneously bound. The paper is ready for publication, there are two minor points that could be addressed in the final version:

1. The fraction f, as introduced in the new section on the continuum model, represents the fraction of simultaneous binding for both antibodies. Fig. 4A shows 1-f for the antibodies in the study. I find it striking that the values on the diagonal are not all at 100%, meaning that the antibodies do not completely block themselves. Could the authors comment on this interesting observation?

2. There appears to be a minor error in the SI, section C.1: Before Eq. S13, it is stated that "It has been proposed that neutralisation is a sigmoidal function of neutralisation", which probably should have been the number of IgG bound.

Reviewer #3: The authors have addressed all my questions esp. regarding the synergistic effect and have carried out additional computation to generalize the model to be applicable for a continuum of competition strength. I can now support the publication of this manuscript.

**Have all data underlying the figures and results presented in the manuscript been provided?**

Reviewer #1: Yes

Reviewer #2: None

Reviewer #3: Yes

PLOS authors have the option to publish the peer review history of their article (what does this mean?). If published, this will include your full peer review and any attached files.

Reviewer #1: No

Reviewer #2: Yes: Michael Lässig

Reviewer #3: No

---

## [Editor Report · Acceptance letter]

20 Apr 2020

PCOMPBIOL-D-19-01827R1 

When two are better than one: Modeling the mechanisms of antibody mixtures

Dear Dr Bloom,

I am pleased to inform you that your manuscript has been formally accepted for publication in PLOS Computational Biology. Your manuscript is now with our production department and you will be notified of the publication date in due course.

With kind regards,

Laura Mallard
